# Enhanced Convolutional-Neural-Network Architecture for Crop Classification

**Mónica Y. Moreno-Revelo** [1,*], **Lorena Guachi-Guachi** [2,*], **Juan Bernardo Gómez-Mendoza** [3,*], **Javier Revelo-Fuelagán** [4,*] and **Diego H. Peluffo-Ordóñez** [1,5,6]

1 Faculty of Engineering, Department of Electronics, Corporación Universitaria Autónoma de Nariño, Pasto 520001, Colombia; peluffo.diego@um6p.ma
2 Department of Mechatronics, Universidad Internacional del Ecuador, Av. Simon Bolivar, Quito 170411, Ecuador
3 Department of Electrical, Electronic and Computing Engineering, Universidad Nacional de Colombia, Sede Manizales, Manizales 170001, Colombia
4 Department of Electronic Engineering, Universidad de Nariño, Pasto 520001, Colombia
5 Smart Data Analysis Systems Group (SDAS Research Group), Ben Guerir 47963, Morocco
6 Modeling, Simulation and Data Analysis (MSDA) Research Program, Mohammed VI Polytechnic University, Ben Guerir 47963, Morocco
* Correspondence: monica.moreno@aunar.edu.co (M.Y.M.-R.); loguachigu@uide.edu.ec (L.G.-G.); jbgomezm@unal.edu.co (J.B.G.-M.); javierrevelof@udenar.edu.co (J.R.-F.)

**Abstract:** Automatic crop identification and monitoring is a key element in enhancing food production processes as well as diminishing the related environmental impact. Although several efficient deep learning techniques have emerged in the field of multispectral imagery analysis, the crop classification problem still needs more accurate solutions. This work introduces a competitive methodology for crop classification from multispectral satellite imagery mainly using an enhanced 2D convolutional neural network (2D-CNN) designed at a smaller-scale architecture, as well as a novel post-processing step. The proposed methodology contains four steps: image stacking, patch extraction, classification model design (based on a 2D-CNN architecture), and post-processing. First, the images are stacked to increase the number of features. Second, the input images are split into patches and fed into the 2D-CNN model. Then, the 2D-CNN model is constructed within a small-scale framework, and properly trained to recognize 10 different types of crops. Finally, a post-processing step is performed in order to reduce the classification error caused by lower-spatial-resolution images. Experiments were carried over the so-named Campo Verde database, which consists of a set of satellite images captured by Landsat and Sentinel satellites from the municipality of Campo Verde, Brazil. In contrast to the maximum accuracy values reached by remarkable works reported in the literature (amounting to an overall accuracy of about 81%, a $f_1$ score of 75.89%, and average accuracy of 73.35% ), the proposed methodology achieves a competitive overall accuracy of 81.20%, a $f_1$ score of 75.89%, and an average accuracy of 88.72% when classifying 10 different crops, while ensuring an adequate trade-off between the number of multiply-accumulate operations (MACs) and accuracy. Furthermore, given its ability to effectively classify patches from two image sequences, this methodology may result appealing for other real-world applications, such as the classification of urban materials.

**Keywords:** convolutional neural network (CNN); crop classification; post-processing; satellite images

## 1. Introduction

Agriculture is one of the main economic activities in the world. Today, considering the continuous human growth population and the limited food availability, agricultural activities need to be monitored on regular basis, in such a manner that the increase in efficiency in food production is enabled, while protecting the natural ecosystems [1–4]. In this context, crop classification can be used to provide information about production and thus becomes a useful tool for developing sustainable plans and reducing environmental

issues associated with agriculture [5–7]. As a result, timely collection and the analysis of data from large crop areas is of great interest. Traditionally, such analysis is carried out by using computational tools and satellite imagery processing with artificial intelligence (AI) techniques [8–11].

Throughout the years, several AI techniques have been explored to tackle the problems related with crop classification [12,13]. In such vein, machine learning (ML) benchmark algorithms have been successfully used from both unsupervised [14,15] and supervised [16] inferences. Nonetheless, conventional ML techniques may not be recommendable (and may even be prohibitive) when a manual feature extraction stage is unfeasible. In addition, ML approaches may require exhaustive parameter tuning to reach a high accuracy. In this connection, by overcoming these drawbacks, deep learning (DL) has recently taken place as one of the most appealing approaches. Broadly, DL approaches can be divided into artificial neural networks (ANNs), recurrent neural network (RNNs) and convolutional neural networks (CNNs) [17]. Particularly, for image classification problems, CNNs and RNNs are preferred over conventional ANNs as they extract sequential information. RNNs have proven to be effective at extracting temporal correlations and classifying data as a whole while maintaining a manageable computational complexity. In this regard, some works [18,19] have proposed novel network architectures based on RNNs combined with CNNs for automated feature extraction from multiple satellite images through learning time correlation.

CNNs are of special interest as their main advantage over other techniques—besides automatically extracting features through convolutional layers—is their ability to capture spatial features (i.e., the pixel arrangement and relationships thereof) [20,21] as well as its versatility [22,23]. Recent works based on hybrid methods combining 2D- and 3D-CNN [24] have proven that 3D-CNNs enable the joint spatial–spectral feature representation from stacked spatial bands. Such methods have been shown to be less computationally expensive than those solely based on 3D-CNN architectures [25–27]. In addition, some exploratory studies exhibit that 3D-CNNs may underperform 2D-CNNs when classes have similar textures across multiple spectral bands [28], and therefore 2D-CNN-based approaches are preferred by the vast majority of recent research works on crop classification. Following from these insights and given that this work is not intended to include spatial–temporal information, 2D-CNNs are the architectures of choice for this study.

Nonetheless, despite being appealing for image analysis in agriculture settings, techniques based on 2D-CNNs may involve complex architectures with a great amount of layers and parameters entailing a high computational cost and training time [25,29]. Therefore, there is still a need for creating more accurate crop classification systems which involve a lower computational burden.

Aimed at establishing a way forward to provide solutions in this regard, this work introduces a competitive methodology for crop classification from multispectral satellite imagery by taking advantage of using an enhanced 2D-CNN together with a novel post-processing step. Inspired by the main workflow of various RNN/2D-CNN-based methods for patch classification [20,30–34], the proposed methodology mainly contains four steps: image stacking, patch extraction, 2D-CNN based modelling for classification purposes and post-processing, as depicted in Figure 1.

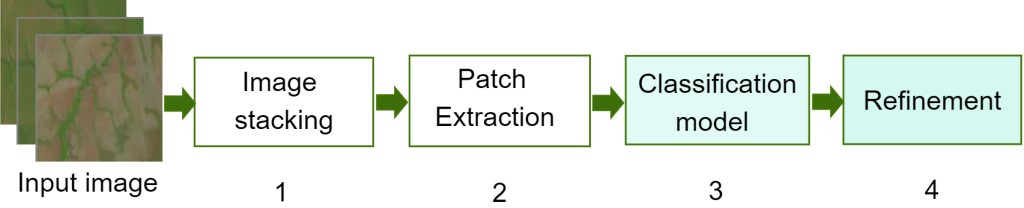

**Figure 1.** The proposed methodology's high-level scheme: the benefit of using an enhanced 2D-CNN as a classification model alongside with a post-processing stage for map refinement is highlighted.

In broad terms, the proposed methodology works as follows: (1) the images are stacked to increase the number of features; (2) the input images are split into patches and fed into the 2D-CNN; (3) the 2D-CNN architecture is designed following a smaller-scale approach, and trained to recognize 10 different types of crops (soybean, corn, cotton, sorghum, non-commercial crops (NCC), pasture, eucalyptus, grass, soil, and Cerrado); and (4) reduce the classification error caused by lower-spatial-resolution images, and a here-introduced post-processing step is performed.

Images used in this work are from the Campo Verde database, introduced in [30], which corresponds to a Landsat 8 imagery of a tropical region (Municipality of Campo Verde) from Brazil. Its analysis represents a challenging problem as it holds a wide range of crops, and related research works have mostly been devoted to studying non-tropical areas.

As for the design and implementation of the enhanced 2D-CNN, the following architecture is proposed: firstly, the 2D-CNN is trained independently for each one of the sequences by extracting patches of $32 \times 32 \times n$ pixels in size, where $n$ is the number of bands in the sequence. Secondly, the training patches are passed through three layers of convolution, yielding the feature maps. Thirdly, a pooling layer reduces the feature map size. Finally, the classification task itself is carried out by a fully-connected layer and the output layer.

As for the post-processing stage, it can be said that it consists in refining the annotations obtained by the 2D-CNN by eliminating discontinuities and misclassified pixels through morphological operators. This post-processing becomes crucial as it makes the the methodology more robust to lower-spatial-resolution images and therefore improves the classification rate.

The experimental framework used in this research is designed by following those developed in similar works [30,31]. Two sequences of the Campo Verde database are considered: 1. from October 2015 to February 2016; and 2. from March to July 2016. The proposed methodology achieves a $f_1$ score of 75.89%, which is higher than the previous results reported in the literature [20,30–34]. Indeed, a competitive rise of the classification rate in contrast to benchmark-and-recent works conducting experiments on the same dataset was accomplished. This performance is attributed to the exhaustive search of parameters across all the stages of the proposed methodology (namely, the selection of patches size, and setting of the number of filters for the CNN). For comparison purposes, the ability of an RNN architecture alone to classify patches as a whole while maintaining low computational complexity is also explored. Additionally, a biological technique (iterative label refinement) is also evaluated to make comparisons with the proposed post-processing.

This paper is structured as follows: Section 2 presents a brief review of state-of-the-art related works. The database and the methods of the proposed methodology's building blocks are described in Section 3. Section 4 describes the experiments carried out over the Campo Verde database. Section 5 presents the results and discussion across the experiments. Some additional results of evaluating the proposed methodology over urban-materials-related images (Pavia scenes) are presented in Section 6. Finally, Section 7 gathers the concluding remarks.

## 2. Related Works

Satellite and aerial images have been classified with several techniques, including ML and DL methods for different purposes. A recent survey about different techniques used in remote sensing [17] reports that DL architectures is one of the most used techniques for farming applications. Therefore, remarkable studies on crop classification from images have been devoted to this kind of technique. The authors in [35] carried out a study on remotely sensed time series in California in order to classify 13 summer crop categories. The authors compared traditional methods such as random forest and support vector machine (SVM) with two DL architectures: a long short-term memory (LSTM) and a 1D-CNN, demonstrating that the 1D-CNN architecture reaches the best results (an accuracy around 85.54%) among all DL and non-deep learning models. Similarly, Ref. [36] is devoted

to classify 14 types of crops in Sentinel images along 254 hectares of land in Denmark. It used a DL architecture inspired by the results from the combination of two networks, a fully connected network (FCN) and an RNN. The latter architecture was also used by [37] for classifying crops in Sentinel images of France; the classified crops are rice, sunflower, lawn, irrigated grassland, durum wheat, alfalfa, tomato, melon, clover, swamps, and vineyard. The proposed approach is compared against traditional methods such as the *k*-nearest neighbor (*k*-NN) and SVM, exhibiting better results when using the DL approach.

The work carried out in [38] classifies 22 different crops of aerial images using a CNN with local histograms. The method extracts information related with texture patterns and color distribution to achieve scores of 90%. The study in [18] combines a CNN and an RNN in a pixel-based approach to classify 15 types of crops on multi-temporal Sentinel-2 imagery of Italy. The DL method was compared with ML methods including SVM and RF. The best accuracy values were reached by the R-CNN approach with a value of 96.5%. A novel classification technique is proposed in [19], which applies transfer learning (TL) to solve the problem related to imbalanced databases. It was tested on a crop database with the aim of recognizing pests. Furthermore, this research compares various CNN architectures and achieves an accuracy over 95%. Another work [39] uses an RNN-based approach to classify SAR data from China.

More specialized studies have explored the benefit of using 3D-CNNs in spatio-temporal remote sensing images. For instance, in [25], a new paradigm for crop identification area by using 3D-CNNs is introduced to incorporate the dynamics of crop growth. Likewise, the research presented in [26] used a 3D-CNN to classify four scenes where urban areas as well as crops (including lettuce and corn) could be found. The proposed 3D-CNN together with a principal component analysis (PCA) stage (applied to extract the most important information from the images) achieves an overall accuracy above 95%. Another 3D-CNN model for cloud classification was explored in [27], which was tested over two databases (GF-1 WFV validation data and ZY-3 validation data). Such a model reaches an accuracy of 97.27%. The work developed in [40] classifies a tree database of Finland. There are 4142 trees clustered into three classes (pine, spruce and birch). The classification stage was carried out with four convolutional layers and three max pooling layers, accomplishing an accuracy of approximately 94%. A very recent work in the field of hyperspectral image analysis using 3D-CNN [41], similarly to [40], mainly aimed to classifying trees from a Finnish database. It compared both DL and ML methods, and experimentally demonstrated that 3D-CNN yields the best results (above 91% of accuracy). A 3D-CNN is also successfully used to classify soybean and grass from the MODIS data of USA [42]. The preliminary results of a 3D-CNN classification model of cotton and corn crops from the Sentinel data of Turkey are outlined in [43].

A worth-reviewing paper overviewing of applications and advances of DL in agriculture in detail and information on key aspects (such as databases, and typical crop classification problems, among others) is presented in [5]. Another work of great interest is that reported in [44], which outlines CNNs and their applications in different fields.

Regarding the use of the Campo Verde database for crop classification purposes, the following works are worth making note of. The Campo Verde database was introduced in [30], with the intention to outline the problems related to the farming field for educational purposes. It is documented and manually annotated. [30] also depicted an experiment with a random forest classifier. The study in [31] evaluates a CNN and a FCN obtaining greater efficiency when using the latter architecture. For experiments, two sequences consisting of an image stacking were evaluated: the first sequence corresponding to the period from October 2015 to February 2016, and the second sequence from March to July 2016. As a conclusion, the FCN is proven to be a better solution outperforming the baseline in terms of processing time in the inference phase. Another similar technique was proposed in [20], which uses CNN- and RNN-based architectures. This study reported higher results for the RNN since the CNN is fed with the co-occurrence matrix features rather than extracting their own features. A classification approach of tropical crops is presented in [32], which

followed a method that compares auto-encoders, CNNs, and FCNs networks in terms of segmentation performance over Sentinel images of Campo Verde. As a result, it was observed that the accuracy was better when using DL techniques (specifically, CNN), rather than the other considered methods—indeed, the CNN-based approach showed a more stable behavior. The research presented in [33,34] apply DL techniques to classify not only Campo Verde but also Luis Eduardo database [45] (another agriculture database from Brazil). Both works use approaches based on an FCN and similar parameters including a 32 × 32 patch size and assess the methodology over individual images and sequences composed by several images. It is worth mentioning that [34] adds an LSTM structure to the methodology, which allows learning from multi-temporal data. The incorporation of such LSTM yielded a remarkable increase in the overall performance, highlighting then the importance of the temporal information for crop recognition.

At this extent, it is worth noticing that most of the previously mentioned works were traditionally tested only on Sentinel images of Campo Verde (not the Landsat images) for crop classification tasks since Landsat images are covered by clouds. Furthermore, studies have accomplished a $f_1$ score not over 75%. Likewise, in [46], it is highlighted that the classification of heterogeneous crops is still a challenging open issue as there exists a diverse and complex range of spectral profiles of crops, as it is the case of Campo Verde.

## 3. Materials and Methods

### 3.1. Database

This work uses the public Campo Verde database [30], which is a satellite imagery of crops from the municipality of Campo Verde, Brazil. It is of great interest as it is a tropical area. Figure 2 shows the class (types of crop) occurrences per image in the dataset. Notice that the crop of beans are not included as the image sequences used in this work contain no pixels of this crop.

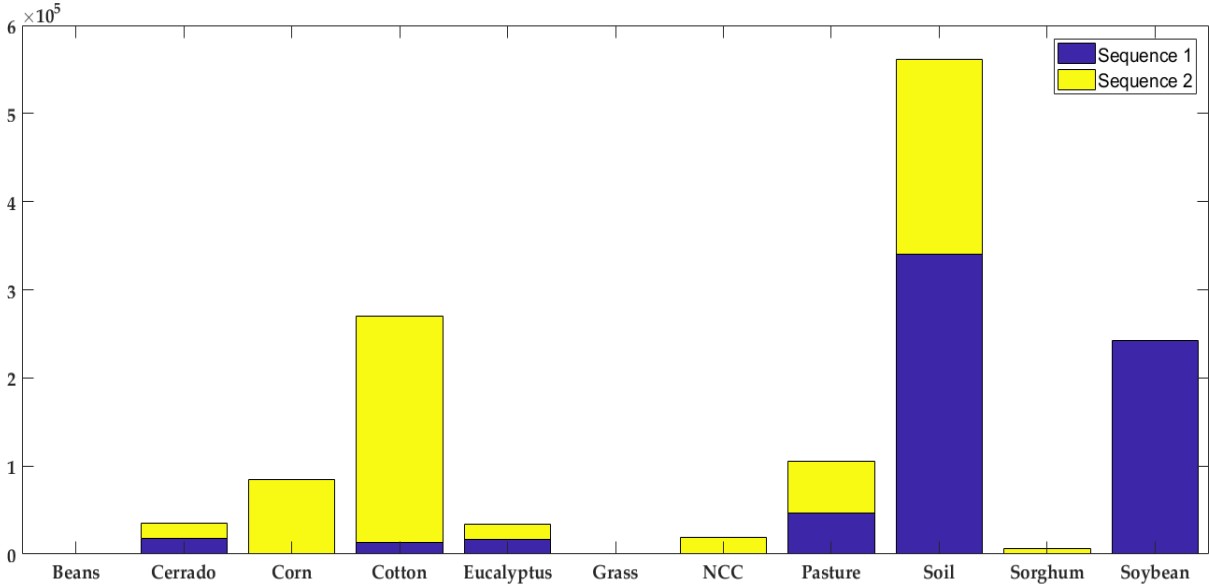

**Figure 2.** Distribution of the selected classes (soybean, corn, cotton, sorghum, non-commercial crops (NCCs), pasture, eucalyptus, grass, soil, and Cerrado) per image sequence from Campo Verde database.

Campo verde consists of a set of 14 Sentinel-1 images, a set of 16 Landsat 8 OLI images, and an ESRI Shapefile containing the daily class annotations in each and every area of the images. For further experiments, images corresponding to the time between October 2015 and July 2016 (16 Landsat images) are considered.

To ensure that the obtained results become comparable with those of [30,31], images covered by clouds in the Landsat are excluded (two images from January, one from

May, one from June, and one from July). As a matter of fact, the experiments in [30,31] are carried out over Sentinel images on the same date and having no affectation by cloud cover.

Specifically, the number of classes for sequences 1 and 2 was set as 9 and 10, respectively. Selected classes (namely: soybean, corn, cotton, sorghum, non-commercial crops (NCC), pasture, eucalyptus, grass, soil, and Cerrado) correspond to the annotations of the last image from each sequence.

The bean class is contained in other months including October but not in the months analyzed in the sequences (February and July). Moreover, the classes of crops become represented in an imbalanced way as samples from some crops greatly outnumber the ones from other crops.

### 3.2. Proposed Methodology

The proposed methodology aims at improving the accuracy obtained by outstanding state-of-the-art methods for crop classification tasks. The methodology is composed by four main phases as shown in Figure 3, where $n$ is the number of bands of the image and $P$ is the patch size. Firstly, the satellite images are stacked to increase the number of features. Second, the stacked images are divided into small patches to analyze local-feature details. Third, the patch classification process is performed by a DL model. Finally, a post-processing phase is added to mitigate the misclassified pixels introduced due to the low spatial resolution of the images.

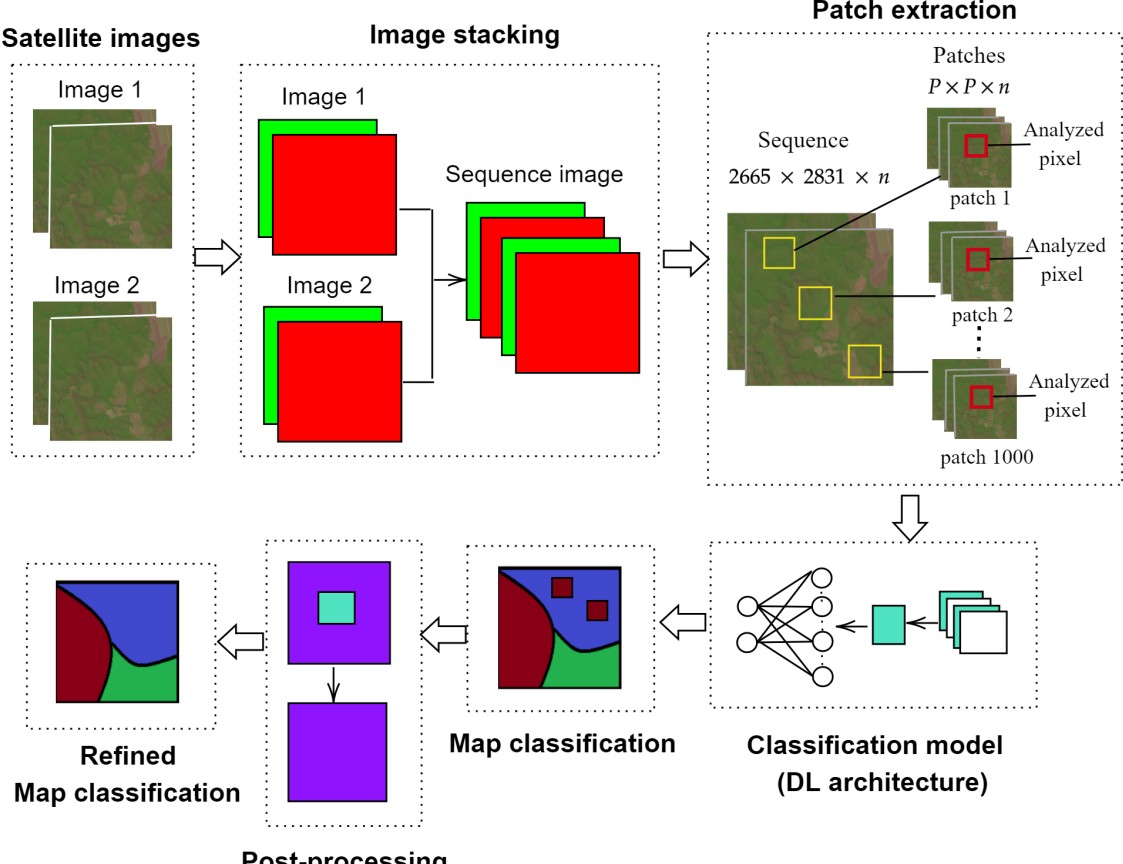

**Figure 3.** Work-flow of the proposed methodology for crop classification from satellite images. This mainly involves stages for image stacking, patch extraction, classification model, and post-processing.

### 3.2.1. Image Stacking

In this stage, two images are piled to augment the number of features analyzed for each pixel and thus improve the overall performance for crop classification. This procedure is reported in [30]. More concisely, for the proposed methodology, the images are stacked in only two sequences as follows: the first one was formed by the images from October 2015 to February 2016 (a total of five images), and the second one was formed by the images from March to July 2016 (a total of six images). Each Landsat image has seven bands, consequently, sequences 1 and 2 yield images with 35 and 42 bands, respectively.

### 3.2.2. Patch Extraction

This stage generates a set of valid patches for the 2D-CNN and RNN of size $32 \times 32 \times n$ and $8 \times 8 \times n$, respectively, centered on each pixel to classify the central pixel based on neighborhood frequency information, where $n$ is the number of bands of each sequence. In order to obtain a suitable trade-off among computational cost, context capturing, and localization accuracy, a small sample size is used. At this stage, each patch is examined, and the frequency of pixels with similar class values to the pixel of interest (central pixel) is determined. When the frequency is greater than 512 pixels, the patch is selected as a valid patch, otherwise the patch is discarded. Then, for each class, 1K valid patches are randomly selected. The outcome of this stage is a training set of 10K for sequences 1 and 9K for sequence 2. Since corn, sorghum, bean, and soil are scarce in February and July, the number of patches is less than 1K. Therefore, to handle this issue, synthetic fields for these crops are developed using geometric transformations such as horizontal ($T_x$) and vertical ($T_y$) translations with 1 *cm* displacements on the available data. Such geometric transformations are benchmarking data augmentation methods that have effectively augmented the original data in some DL applications [47].

The geometric transformations applied are given by Equation (1):

$$\begin{bmatrix} x' \\ y' \\ 1 \end{bmatrix} = \begin{bmatrix} 1 & 0 & T_x \\ 0 & 1 & T_y \\ 0 & 0 & 1 \end{bmatrix} \begin{bmatrix} x \\ y \\ 1 \end{bmatrix}, \tag{1}$$

where $x'$ and $y'$ are the transformed coordinates and $x$ and $y$ are the initial coordinates of the pixels; $T_x$ and $T_y$ are the values of the horizontal and vertical translations. Table 1 summarizes the amount of synthetic patches generated for each class used in this work.

**Table 1.** Amount of synthetic patches created per selected classes (soybean, corn, cotton, sorghum, non-commercial crops (NCC), pasture, eucalyptus, grass, soil, and Cerrado) by using horizontal and vertical translations.

| Crop | Soybean | Corn | Cotton | Sorghum | NCC | Pasture | Eucalyptus | Grass | Soil | Cerrado |
|---|---|---|---|---|---|---|---|---|---|---|
| **Sequence 1** | 0 | 576 | 0 | 800 | 832 | 0 | 0 | 900 | 0 | 0 |
| **Sequence 2** | - | 0 | 0 | 0 | 0 | 0 | 0 | 900 | 0 | 0 |

### 3.2.3. Classification Model

Both the accuracy and the increase in the number of parameters to be learned are influenced by the architecture depth. Moreover, the number of parameters has a direct relationship with the computational cost. Therefore, the deeper the architecture is, a larger number of varied samples to build robust models and prevent over-fitting is needed. To tackle this issue, this work proposes enhanced DL architectures for crop classification based on patch analysis. The studied DL architectures are described in Table 2. Considered architectures are intended to achieve high accuracy at low-depth while being able to deeply and meaningfully learn features from stacked multi-spectral images. Particularly, the DL architectures considered in this work are 2D-CNN type. Additionally, an RNN architecture was also evaluated. They were compared to select the most suitable DL model for crop classification.

**Table 2.** DL architectures settings.

| 2D-CNN 1 | 2D-CNN 2 | RNN |
| --- | --- | --- |
| **Architecture** | **Architecture** | **Architecture** |
| Input: $n$ patches size $32 \times 32$ | Input: $n$ patches size $32 \times 32$ | Input: $n$ patches size $8 \times 8$ |
| Conv1: 32 filters size $2 \times 2$ | Conv1: 60 filters size $5 \times 5$ | Memory layer: 32 neurons |
| Conv2: 64 filters size $2 \times 2$ | MaxPooling: size $2 \times 2$ | Hidden layer 1: 64 neurons |
| Conv3: 128 filters size $2 \times 2$ | Conv2: 70 filters size $3 \times 3$ | Hidden layer 2: 100 neurons |
| MaxPooling: size $8 \times 8$ | MaxPooling: size $2 \times 2$ | Output layer: $m$ classes |
| Hidden layer: 100 neurons | Conv3: 80 filters size $1 \times 1$ | |
| Output layer: $m$ classes | Hidden layer: 100 neurons | |
| | Output layer: $m$ classes | |
| **Loss function** | **Loss function** | **Loss function** |
| Cross entropy | Cross entropy | Cross entropy |
| **Activation** | **Activation** | **Activation** |
| Softmax in the output layer | Softmax in the output layer | Softmax in the output layer |
| Relu in the other layers | Relu in the other layers | Relu in the other layers |
| **Weight initialization** | **Weight initialization** | **Weight initialization** |
| Randomly | Randomly | Randomly |
| **Drop out layer** | **Drop out layer** | **Drop out layer** |
| Dropout 25% after pooling | Dropout 50% after hidden layer | Dropout 50% after first hidden layer |
| Dropout 50% after hidden layer | | |

2D-CNN Model

A 2D-CNN is an architecture composed of a sequence of convolutional and pooling layers mostly used to learn features from images [48,49]. This kind of architecture often ends with fully-connected layers to predict a single class label or a set of class probabilities [50–52]. Convolutional layers apply filters over all pixels of the input image to obtain a set of high-level abstract features; pooling layers reduce the features number—controlling the over-fitting; and fully-connected layers reshape the output into a vector with a size equal to the number of classes [53].

For classification purposes, a 2D-CNN commonly applies two activation functions: `softmax` for the output layer and rectified linear unit (`ReLu`) for the rest of the layers. `Softmax` aims at scaling the outputs between zero and one, providing a probability of belonging of coverage to a specific class. `ReLu` is a linear function that will output the input directly if it is positive. Otherwise, it will output zero [54,55].

Furthermore, 2D-CNN 1 is composed of three successive convolutional layers followed by a max-pooling layer. For this sequence, the filter size ($fs$) was chosen among $fs = [2 \times 2, 4 \times 4, 8 \times 8]$ being the first value that allowed to achieve the highest performance. Figure 4 depicts a graphical explanation of the architecture 2D-CNN 1.

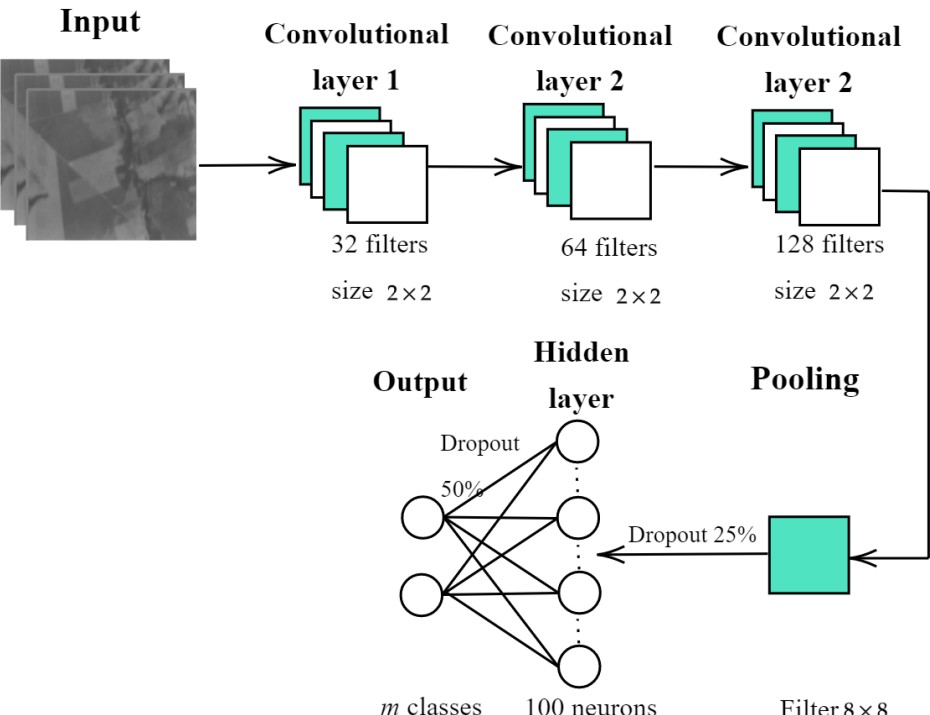

**Figure 4.** A schematic view of the proposed 2D-CNN 1.

On the other hand, the 2D-CNN 2 is composed of three convolutional layers, each of which is followed by a max-pooling of $2 \times 2$. In this architecture, a $1 \times 1$ convolutional layer is applied in order to extract more features of the images without losing information. Then, dropout layers are applied to deactivate a percentage of the neurons and prevent overfitting. Figure 5 shows an explaining diagram of the architecture 2D-CNN 2.

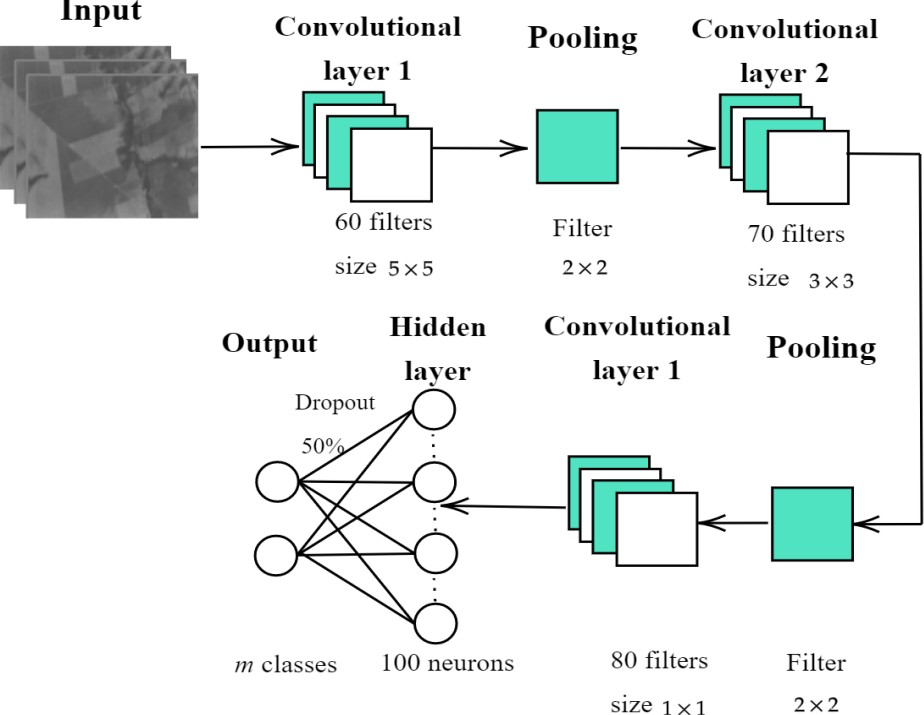

**Figure 5.** A schematic view of the proposed 2D-CNN 2.

RNN Model

An RNN model for image classification analyzes the image as a single sequence of pixel vectors. RNN has neurons in its architecture known as memory cells, which create a notion of temporality [36]. Contrary to the 2D-CNN, before feeding the network, the multi-spectral images are reshaped as spectral signatures since there are only hidden layers in the RNN. For this work, its architecture is composed of one memory layer of 32 neurons followed by two hidden layers (one with 64, and another one with 100 neurons).

### 3.2.4. Map Classification

After training DL models, the resulting models are used to classify each pixel from satellite imagery, where the perceptual field (fov) for each image is assumed as a sliding window of the input size expected by each DL model (see Figure 6). This process ended when all pixels are classified.

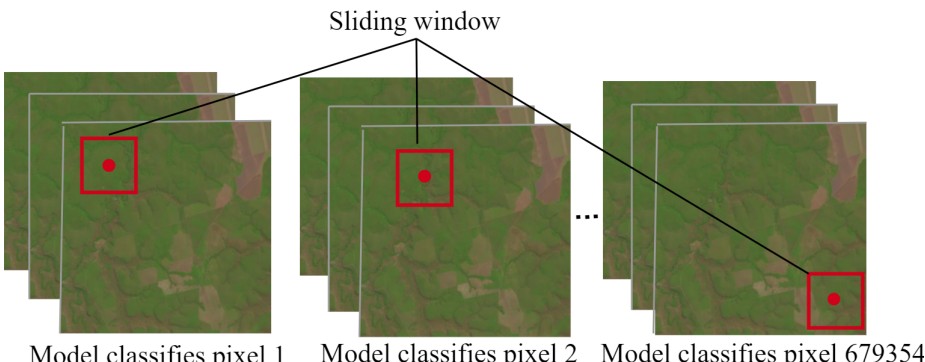

**Figure 6.** Example of crop identification based on patch classification.

### 3.2.5. Post-Processing

Since images from satellites have low spatial resolution, the proposed methodology explores two post-processing techniques to overcome problems of apertures and disconti-nuities in crops with similar features.

Post-Processing Based on Morphological Operations (PMO)

This technique analyzes the map classified by a sliding window of $3 \times 3$. If the prominent class has more than six pixels in the window and it is different from the central pixel's class, the central pixel adopts the label of the most predominant class; otherwise, the label value of the central pixel is retained. This technique allows to improve the overall results by reducing small misclassified zones. An example of this technique for the first three iterations is depicted in Figure 7; nevertheless, it is repeated until the entire image is analyzed.

Post-Processing Based on Iterative Label Refinement (PILR)

This technique is a biologically inspired method. It uses a set of color classifiers that convert visual color information from the scene into segmentation labels [56]. This tech-nique is used to remove jagged edges, small holes, and unconnected regions that are common in map classification. The scheme of this technique is illustrated in Figure 8. It performs the linear operation given by Equation (2) with a Gaussian filter of $3 \times 3$ window size, as follows:

$$P^{r+1} = w(\alpha P^0 + (1 - \alpha)(P^r * G_\vartheta)), \tag{2}$$

where $P^0$ is the label map with a size of $R \times C \times O$, with $R$ as the number of image rows, and $C$ as the number of image columns. $P^{r+1}$ y $P^r$ are the label maps in iterations $r + 1$ and $r$. $G_\vartheta$ is a Gaussian window. $\alpha$ establishes the influence of the initial classification map and

$w$ is a non-linear operator which determines the new label or refined label. Each pixel of the label map is represented by a column vector whose values are different for each class.

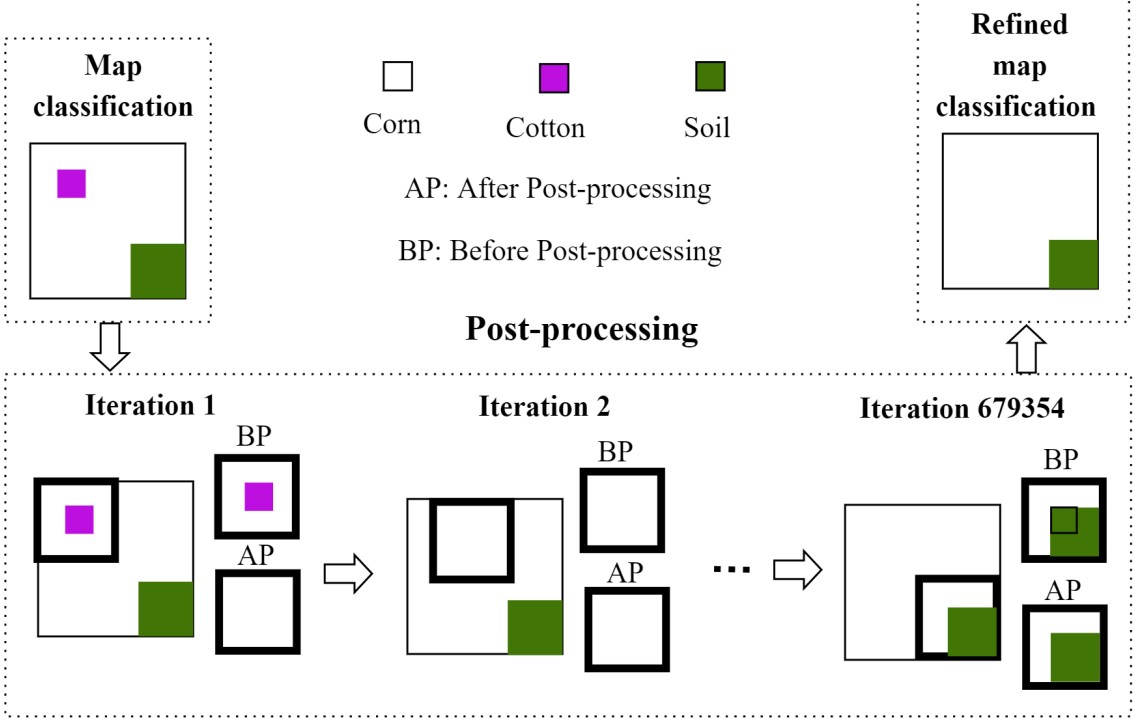

**Figure 7.** Example of the post-processing technique based on consecutive morphological operations.

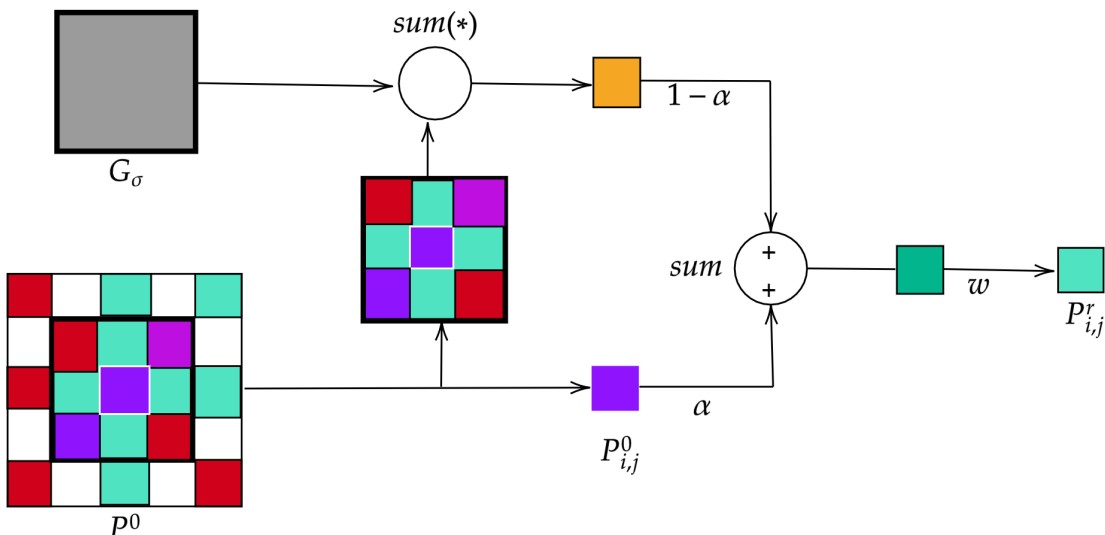

**Figure 8.** Scheme of the post-processing technique based on iterative label refinement.

## 4. Experimental Setup

Since optimal parameter values differ depending on the training samples, in order to carry out a comparative evaluation, the proposed methodology and the evolved methods have been tailored with parameters capable of achieving the highest accuracy to classify accurately crops from satellite images. Python 3.5 (including Python libraries scikit-image, and Tensorflow with Keras) software routines have been implemented to test the proposed

methodology [57,58]. Experiments are done by running the software on an Intel Core i7 processor with 16GB of RAM memory.

### 4.1. Training, Validation and Testing

For learning the DL architecture weights, this work uses the stochastic gradient descent algorithm, Adadelta optimizer with a batch size size of 20, and an entropy cost function[59]. The patches dataset was divided into two sets with a percentage ratio of 90% and 10% for training and validation, respectively (in a cross-validation scheme, using 10 randomly selected partitions). Table 3 provides basic information about the optimal parameter values of all evaluated DL models, including the parameter's name with corresponding meaning and their experimentally selected values. CNN model parameters are in line with values reported in [30].

**Table 3.** Parameter values for all experiments.

| Parameter | Sequence 1 | Sequence 2 | Description |
|---|---|---|---|
| Patch size | $32 \times 32 \times 35$ | $32 \times 32 \times 42$ | Input shape for feeding 2D-CNN architectures |
| Patch size | $8 \times 8 \times 35$ | $8 \times 8 \times 42$ | Input shape for feeding RNN architecture |
| Training samples | 9 K | 8.1 K | Number of patches to train the DL architectures |
| Validation samples | 1 K | 900 | Number of patches to evaluate the DL architectures |
| Testing samples | 6.69 K | 6.70 K | Number of patches to test the DL architectures |
| Learning rate | 32 | 20 | Control how much to change the model |
| Optimizer | Adadelta | Adadelta | Fit weights |
| Cost function | Entropy | Entropy | Estimates the error |
| Metric | Accuracy | Accuracy | Evaluates the models |
| Activation function 1 | Relu | Relu | Establish the output in hidden layers |
| Activation function 2 | Softmax | Softmax | Establish the o |
| Batch size | 20 | 32 | Number of samples per training iteration |
| Epoch | 30 | 30 | Number of passes over the whole dataset |
| Iterations | 450 | 330 | Maximum number of iterations |

It should be noted that the overall performance of the proposed methodology is evaluated on 1.3M samples to obtain fair comparison results with the works reported in [30,31].

### 4.2. Performance Measurements

The overall accuracy ($OA$), average accuracy ($AA$) and $f_1$ score ($f_1$) given by Equations (3) and ( are the most widely used metrics for evaluating the ability of classifying images into corresponding class.

$OA$ quantifies the ratio of correctly classified samples. $AA$ is the average of each accuracy per class, and $f_1$ quantifies the harmonic mean between precision and recall. $f_1$ is often used when the false negatives and false positives are crucial.

Precision computes the number of positive predictions divided by the total number of positive class values predicted, whereas recall determines the number of positive predictions divided by the number of positive class values in the test dataset. Following are their corresponding formulas:

$$OA = \frac{TN + TP}{TP + TN + FP + FN}, \tag{3}$$

$$f_1 = \frac{2 \times \text{precision} \times \text{recall}}{\text{precision} + \text{recall}}, \text{ with} \tag{4}$$

$$\text{recall} = \frac{TP}{TP + FN}, \text{ and} \tag{5}$$

$$\text{precision} = \frac{TP}{TP + FP}, \tag{6}$$

where $TP$ is the number of patches that are predicted as positive by the model that are actually positive, $FP$ is the number of patches that are labeled as positive by the model but are actually negative. The number of patches predicted as negative by the model but actually are positive is $FN$, while the number of patches marked as negative by the model but actually negative is $TN$, as shown in confusion matrix (see Figure 9).

| | | Prediction | |
|---|---|---|---|
| | | **Positives** | **Negatives** |
| **Label** | **Positives** | True Positives ($TP$) | False Negatives ($FN$) |
| | **Negatives** | False Positives ($FP$) | True Negatives ($TN$) |

**Figure 9.** General explanation of confusion matrix.

The computational complexities of all the compared DL architectures are evaluated in terms of the number of multiply–accumulate operations ($MACs$) required in their main classification step. In a fully connected layer, all the inputs are connected to all the outputs. Thus, for a layer with $I$ input values and $J$ output values, its weights $W$ can be stored and computed in the matrix given by Equation (7):

$$MACf_{cl} = I \times J. \tag{7}$$

Since the convolutional layers are three-dimensional feature maps of size $H \times W \times C$ where $H$ is the height of the feature map, $W$ the width, and $C$ the number of channels at each location, for a convolutional layer with kernel size $K$, the number of $MACs$ is given by Equation (8):

$$MAC_{cl} = K \times K \times C_{in} \times H_{out} \times W_{out} \times C_{out}. \tag{8}$$

## 5. Results and Discussion

This section reports three sets of experiments derived from the scheme shown in Figure 3, which examine the impact of the introduced DL architectures and the post-processing phase on the overall performance of the proposed methodology to accurately classify pixels from satellite images in a sliding window fashion.

### 5.1. Comparison between Results of DL Architectures

The proposed 2D-CNNs and RNN architectures are compared to determine their ability to correctly classify image patches. Table 4 reports $OA$, $AA$, and $f_1$ achieved by each architecture.

It can be observed that 2D-CNN 1, characterized by three consecutive convolutional layers, leads to higher $OA$ evaluated architectures. However, it is worth noting that the achieved score for sequences 1 and 2 is only 2.54% and 0.38%, higher than that reached by 2D-CNN 2, respectively. This is explained since it down-sample features once before applying fully connected layers resulting in a model with the presence of a greater quantity of features that allow to classify crops correctly.

These results also indicate that convolutional layers followed by a pooling layer in 2D-CNN 2 might down-sample useful features for purposes of this work. Furthermore, the remarkable accuracy, higher than 12%, achieved by 2D-CNN 1, and 2 over RNN architecture was also observed. The average $AA$ achieved by the CNN in sequence 1, for example, is strongly higher than the value achieved for the RNN.

Also it is demonstrated that the features obtained by the convolutions applied over the images allow to obtain higher accuracy than the spectral signature features used by the RNN. Since 2D-CNN 1 will ensure a significantly high $OA$, $AA$, and $f_1$, it has been chosen as the best trade-off for the proposed methodology and used in the map classification phase.

**Table 4.** The overall performance of the proposed DL architectures for patch classification tasks in terms of *OA*, *AA*, and *f*₁ metrics. The best measurements achieved by each architecture are highlighted and it is observed that the 2D-CNN 1 leads to higher *OA* values in both image sequences.

| Sequence | Class | # of Pixels | 2D-CNN 1 | | | 2D-CNN 2 | | | RNN | | |
|---|---|---|---|---|---|---|---|---|---|---|---|
| | | | $f_1$(%) | $AA$(%) | $OA$(%) | $f_1$(%) | $AA$(%) | $OA$(%) | $f_1$(%) | $AA$(%) | $OA$(%) |
| | Soybean | 242 K | 79.76 | 76.95 | | 80.03 | 82.72 | | 70.67 | 66.37 | |
| | Corn | 891 | 77.91 | 87.65 | | 72.22 | 82.71 | | 18.61 | 95.17 | |
| | Cotton | 12 K | 78.22 | 96.13 | | 81.65 | 94.37 | | 34.52 | 90.69 | |
| | Sorghum | 242 | 59.99 | 100 | | 57.96 | 100 | | 24.31 | 85.95 | |
| Sequence 1 | NCC | 892 | 61.11 | 99.77 | 84.26 | 66.12 | 98.99 | 81.72 | 18.68 | 66.92 | 66.39 |
| | Pasture | 46 K | 84.98 | 89.85 | | 77.18 | 84.26 | | 56.56 | 86.87 | |
| | Eucalyptus | 17 K | 92.85 | 92.95 | | 93.31 | 90.88 | | 80.63 | 72.31 | |
| | Soil | 340 K | 87.28 | 87.3 | | 84.01 | 79.06 | | 71.88 | 62.55 | |
| | Grass | 112 | 29.71 | 100 | | 34.78 | 100 | | 63.15 | 80.35 | |
| | Cerrado | 17K | 84.43 | 91.49 | | 72.18 | 93.30 | | 55.73 | 81.94 | |
| Average | - | - | 73.63 | 92.21 | 84.26 | 71.94 | 90.63 | 81.72 | 49.48 | 78.91 | 66.39 |
| | Corn | 84 K | 61.96 | 80.22 | | 59.83 | 69.89 | | 47.81 | 70.33 | |
| | Cotton | 257 K | 87.43 | 94.96 | | 83.99 | 81.61 | | 80.01 | 71.48 | |
| | Sorghum | 6.3 K | 76.56 | 79.83 | | 62.91 | 75.87 | | 45.03 | 94.98 | |
| | NCC | 18 K | 78.63 | 87.63 | | 73.34 | 91.24 | | 63.86 | 88.15 | |
| Sequence 2 | Pasture | 58 K | 89.15 | 89.79 | 77.32 | 82.45 | 84.73 | 76.94 | 65.01 | 82.46 | 64.11 |
| | Eucalyptus | 17 K | 90.49 | 97.77 | | 83.34 | 97.70 | | 85.78 | 90.39 | |
| | Soil | 220 K | 65.54 | 50.01 | | 72.94 | 67.13 | | 52.93 | 40.94 | |
| | Grass | 112 | 65.68 | 100 | | 58.18 | 100 | | 40.01 | 85.71 | |
| | Cerrado | 17 K | 85.26 | 90.46 | | 85.93 | 87.38 | | 66.53 | 94.08 | |
| Average | - | - | 77.86 | 85.63 | 77.32 | 73.66 | 83.95 | 76.94 | 60.78 | 79.84 | 64.11 |

## 5.2. Comparison between the Results of Post-Processing Techniques

To explore the advantage of adding the post-processing phase, this work evaluates two post-processing techniques named: post-processing based on morphological operations (PMO) and post-processing based on iterative label refinement (PILR). By taking average of all $f_1$, $AA$, and $OA$ metrics in Table 5.

It can be seen that PMO reaches the highest accuracy scores, and just 0.04%, 0.02%, and 0.05% higher than $f_1$, $AA$, and $OA$, respectively achieved by the proposed methodology before post-processing (proposed methodology - BP) for sequence 1. However, for sequence 2, PMO is just 0.25%, −0.3%, and 0.66% higher than $f_1$, $AA$, and $OA$, respectively.

On one hand, obtained results demonstrate that the PMO technique slightly improves the overall results achieved by PILR technique. This behavior may be attributed to the distribution of data in the database since the PILR technique removes some minority classes and misclassified them with the class containing the majority samples.

On the other hand, the filter size used in PMO is suitable for the images analyzed because it is small, and consequently, significant information is not removed by the technique.

Thus, although PILR has been demonstrated to be stronger in other works [56], the PMO technique is established in the proposed methodology as a suitable post-processing technique for crop classification. Results for the proposed methodology-BP are also reported in Table 5.

**Table 5.** The overall performance reached by the proposed methodology with post-processing techniques. The most significant differences between the values of the techniques are highlighted. PMO: post-processing technique based on morphological operations. PILR: post-processing based on iterative label refinement; BP: before post-processing.

| Sequence | Class | # of Pixels | PMO | | | PILR | | | BP | | |
|---|---|---|---|---|---|---|---|---|---|---|---|
| | | | $f_1$(%) | $AA$(%) | $OA$(%) | $f_1$(%) | $AA$(%) | $OA$(%) | $f_1$(%) | $AA$(%) | $OA$(%) |
| Sequence 1 | Soybean | 242 K | 79.81 | 76.96 | | 40.47 | 37.71 | | 79.76 | 76.95 | |
| | Corn | 891 | 77.91 | 87.65 | | 24.63 | 20.76 | | 77.91 | 87.65 | |
| | Cotton | 12 K | 78.49 | 96.13 | | 28.97 | 40.63 | | 78.22 | 96.13 | |
| | Sorghum | 242 | 59.97 | 100 | | 8.46 | 17.35 | | 59.99 | 100 | |
| | NCC | 892 | 61.12 | 99.77 | 84.31 | 5.26 | 15.69 | 40.94 | 61.11 | 99.77 | 84.26 |
| | Pasture | 46 K | 85.04 | 89.85 | | 28.27 | 46.34 | | 84.98 | 89.85 | |
| | Eucalyptus | 17 K | 92.85 | 92.95 | | 48.57 | 52.49 | | 92.85 | 92.95 | |
| | Soil | 340 K | 87.34 | 87.47 | | 49.69 | 42.51 | | 87.28 | 87.37 | |
| | Grass | 112 | 29.71 | 100 | | 4.84 | 99.11 | | 29.71 | 100 | |
| | Cerrado | 17 K | 84.43 | 91.49 | | 12.88 | 31.63 | | 84.43 | 91.49 | |
| Average | - | - | 73.67 | 92.23 | 84.31 | 25.2 | 40.42 | 40.94 | 73.63 | 92.21 | 84.26 |
| Sequence 2 | Corn | 84 K | 62.33 | 78.55 | | 29.88 | 43.94 | | 61.96 | 80.22 | |
| | Cotton | 257 K | 87.28 | 95.94 | | 55.19 | 47.86 | | 87.43 | 94.96 | |
| | Sorghum | 6.3 K | 78.17 | 79.83 | | 1.88 | 1.79 | | 76.56 | 79.83 | |
| | NCC | 18 K | 76.96 | 83.62 | | 27.45 | 42.24 | | 78.63 | 87.63 | |
| | Pasture | 58 K | 90.01 | 90.51 | 77.98 | 36.17 | 35.49 | 40.63 | 89.15 | 89.79 | 77.32 |
| | Eucalyptus | 17 K | 90.67 | 97.67 | | 32.79 | 63.79 | | 90.49 | 97.77 | |
| | Soil | 220 K | 65.99 | 50.47 | | 39.07 | 32.36 | | 65.54 | 50.01 | |
| | Grass | 112 | 66.46 | 100 | | 13.56 | 99.11 | | 65.68 | 100 | |
| | Cerrado | 17 K | 84.99 | 90.35 | | 21.17 | 30.31 | | 85.26 | 90.46 | |
| Average | - | - | 78.11 | 85.21 | 77.98 | 28.57 | 44.09 | 40.63 | 77.86 | 85.63 | 77.32 |

### 5.3. Comparison with the State-of-the-Art

Table 6 collects the metric values achieved by all evaluated approaches. The DL approaches described in [20,31–34] and random forest classifiers [30] are selected for purposes of comparison. They have been chosen not only because they are among the most efficient ML approaches in literature, but also because of their similarity to the proposed work, as they perform their crop classification on Campo Verde database.

In addition, for the selected approaches, some parameters, such as the patch size, depth, and filter size must be carefully set to guarantee the best accuracy to be achieved for a given application.

**Table 6.** Overall accuracy from Campo Verde database and comparison with the state-of-the-art. The highest values achieved for each measurement are highlighted.

| Methodology | Image Sequence 1 | | | Image Sequence 2 | | |
|---|---|---|---|---|---|---|
| | $f_1$(%) | $AA$(%) | $OA$(%) | $f_1$(%) | $AA$(%) | $OA$(%) |
| Proposed methodology | 73.67 | 92.23 | 84.31 | 78.11 | 85.21 | 77.98 |
| Proposed methodology - BP | 73.63 | 92.21 | 84.26 | 77.86 | 85.63 | 77.32 |
| FCNN by [34] | ≈70 | - | ≈87 | ≈ 70 | - | ≈75 |
| FCNN by [33] | ≈60 | - | ≈85 | ≈ 60 | - | ≈ 75 |
| FCNN by [32] | ≈60 | - | ≈78 | ≈68 | - | ≈72 |
| RNN-CNN by [20] | - | 64.20 | 77.24 | - | 64.46 | 68.06 |
| Dense-FCNN by [31] | - | 79.1 | 81.7 | - | 67.6 | 74.9 |
| Random Forest [30] | ≈42 | - | ≈65 | ≈52 | - | ≈62 |

Table 7 collects computational complexities data where the number of *MAC* operations is computed for compared DL models where the architecture details are provided.

**Table 7.** Computational complexity of the compared DL models.

| Architecture | Sequences | |
| --- | --- | --- |
| | **Sequence 1** | **Sequence 2** |
| 2D-CNN 1 | **Convolutional layers**<br>Conv1: 4.5 M<br>Conv2: 8.1 M<br>Conv3: 14.1 M | **Convolutional layers**<br>Conv1: 5.5 M<br>Conv2: 9.6 M<br>Conv3: 16.8 M |
| | **Fully connected layers**<br>Hidden layer 1: 3.5 M<br>Output layer: 3.5 K | **Fully connected layers**<br>Hidden layer 1: 4.2 M<br>Output layer: 4.2 K |
| | **Total**<br>30.2 M | **Total**<br>36.2 M |
| | **Prediction time**<br>6 h | **Prediction time**<br>5 h |
| RNN | **Memory layer**<br>Layer1: 1.1 M | **Convolutional layers**<br>Layer1: 1.3 M |
| | **Fully connected layers**<br>Hidden layer 1: 2.1 K<br>Hidden layer 2: 6.4 K<br>Output layer: 1 K | **Fully connected layers**<br>Hidden layer 1: 2.1 K<br>Hidden layer 1: 6.4 K<br>Output layer: 0.9 K |
| | **Total**<br>1.12 M | **Total**<br>1.31 MM |
| | **Prediction time**<br>6 h | **Prediction time**<br>5 h |
| FCNN by [34] | Conv1: 1.5 M<br>Conv2: 1.5 M<br>Conv3: 1.5 M<br>Conv4: 102 K<br>RNN layer: 23.5 M | Conv1: 4.1 M<br>Conv2: 4.1 M<br>Conv3: 4.1 M<br>Conv4: 315 K<br>RNN layer: 66 M |
| | **Total**<br>28.1 M | **Total**<br>78.7 M |
| FCNN by [32] | Conv1: 4.4 M<br>Conv2: 829 K<br>Conv3: 2.5 M<br>Conv4: 102 K | Conv1: 12.4 M<br>Conv2: 2.3 M<br>Conv3: 7.2 M<br>Conv4: 315 K |
| | DB 1: 7.3 M<br>DB 2: 2.5 M<br>DB 3: 184 K<br>DB 4: 737 K | DB 1: 20.6 M<br>DB 2: 7.2 M<br>DB 3: 516 K<br>DB 4: 2.1 M |
| | Sa Mpling 1: 819 K<br>Sa Mpling 2: 286 K | Sa Mpling 1: 2.2 M<br>Sa Mpling 2: 802 K |
| | **Total**<br>19.9 M | **Total**<br>55.7 M |

From Table 6, it can be seen that the proposed methodology with post-processing reaches the highest scores in almost all metrics for image sequences 1 and 2. The closest competitor from the state-of-the-art is represented by the FCNN approach reported in [34] with an $OA \approx 3.31\%$ higher for image sequence 1 and $\approx 2.98\%$ lower for image sequence 2. However, the nice property of the proposed methodology is that, when it did not win, its $OA$ achieved is very close to the highest one.

The major advantage of the proposed methodology over state-of-the-art approaches is the improvement in terms of classification accuracy. This slight improvement is attributed

to the post-processing phase and the exhaustive parameter research carried out during the experiments.

From Table 7, it can be observed that the models with a higher number of *MAC* operations (often deeper models), do not always lead to higher accuracy scores as a proper parameter setting per model is required. For instance, FCNN [34] though being among the most complex models for sequence 2, showed a relatively low *OA* score. On the contrary, in addition to involving the lowest number of *MAC* operations, the proposed 2D-CNN 1 achieves an *OA* score higher than most of the compared DL models.

As it is well known, the CNN depth has a fundamental impact on the overall classification, as well as on the computational resources required for training and deployment purposes. However, the proposed methodology is based on consecutive convolutional layers with a significant reduction in depth regarding the other architectures. For example, in [34] an LSTM and an FCNN are used at the same time to classify Campo Verde, and in [31], 100 filters are used in the convolutional layers. In Table 7, the complexity of the 2D-CNN 1 is compared with the most outstanding architectures used by the works mentioned in Table 6.

### 5.4. Classification Map

Finally, Figure 10 depicts the image sequence 2 classes map obtained with the proposed methodology using 2D-CNN 1 and PMO. Figure 10a describes the true label map and Figure 10b depicts the predicted label map. Graphically, it was observed how some classes were misclassified; for instance, in some regions, cotton was labeled as corn. However, comparing the label maps, it was observed that a good soil classification is achieved.

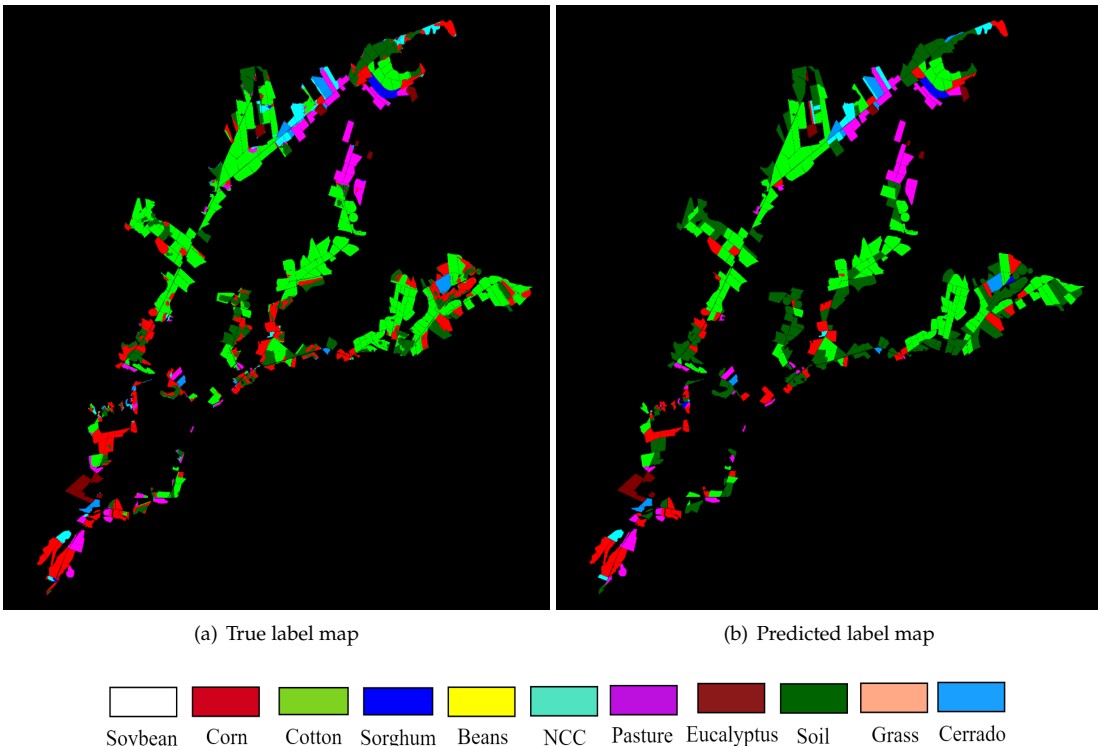

(a) True label map  (b) Predicted label map

Soybean  Corn  Cotton  Sorghum  Beans  NCC  Pasture  Eucalyptus  Soil  Grass  Cerrado

**Figure 10.** Classification map of sequence 2 in Campo Verde database.

## 6. Additional Results: Evaluating Proposed Methodology on Urban Material Classification

To experimentally demonstrate the ability of the proposed methodology to classify images from other domains, some additional experiments were carried out on the Pavia scenes database, which was intended for urban material recognition.

One of its samples is an image acquired by the ROSIS sensor over Pavia University, northern Italy, which has 103 spectral bands, and a size of $610 \times 610$ pixels. Its geometric resolution is 1.3 m, and it is labeled into nine classes (asphalt, meadows, gravel, trees, metal sheets, bare soil, bitumen, bricks, and shadows).

The number of samples for each class is depicted in Figure 11. It was observed that there are more samples labeled as "meadows" while there is just a few numbers of other samples such as shadows and bitumen. Therefore, Pavia scenes classification is a highly imbalanced database.

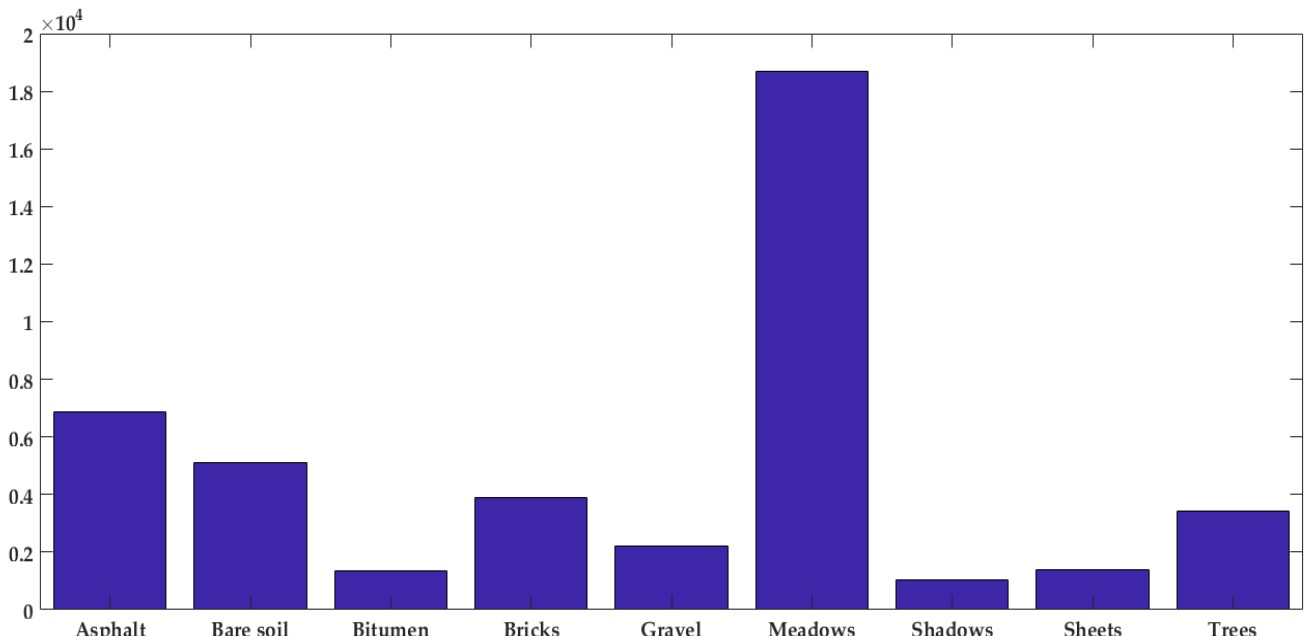

**Figure 11.** Class distribution in Pavia database.

### 6.1. Experimental Setup for Pavia Database

Since the Pavia database contains smaller images, a patch size of $22 \times 22$ was used to classify the Pavia database. It guarantees that each patch has a minimum of 241 pixels of the same class. The analyzed pixel is located at location $(11, 11)$.

The number of patches used for each class is 200, yielding a total number of 1.8K patches. The extracted patches are split into subsets using 90% of the data for training, and 10% of them for validation.

The testing is carried out using the remaining data, i.e., 42,122 pixels from the image (as seen in Table 8).

**Table 8.** Parameter values for experiments applied to Pavia database.

| Parameter | Pavia Image |
|---|---|
| Patch size CNN | $22 \times 22 \times 104$ |
| Total samples | 1.8 K |
| Patch size RNN | $8 \times 8 \times 104$ |
| Training samples | 1.62 K (90 %) |
| Validation samples | 180 (10 %) |
| Testing samples | 4.2 K |
| Learning rate | 32 |
| Optimizer | Adadelta |
| Cost function | Entropy |
| Metric | Accuracy |
| Activation function 1 | Relu |
| Activation function 2 | Softmax |
| Batch size | 32 |
| Epoch | 30 |

### 6.2. Comparison with the State-of-the-Art

Table 9 summarizes the results obtained by the proposed methodology applied to Pavia database which are compared to recent research to classify Pavia database.

It can be seen that the proposed methodology reaches higher $AA$ than [60] while keeping a comparable $OA$.

**Table 9.** Comparison results of the proposed methodology for the Pavia database. The best average values achieved by each method are highlighted.

| Class | # of Pixels | Proposed Methodology | | Proposed Methodology - BP | | RNN | | Chen [60] | |
|---|---|---|---|---|---|---|---|---|---|
| | | *AA*(%) | *OA*(%) | *AA*(%) | *OA*(%) | *AA*(%) | *OA*(%) | *AA*(%) | *OA*(%) |
| Asphalt | 6.8 K | 75.61 | | 75.13 | | 50.55 | | 94.14 | |
| Meadows | 18 K | 93.75 | | 92.56 | | 43.73 | | 92.78 | |
| Gravel | 2.2 K | 75.94 | | 73.17 | | 50.97 | | 80.6 | |
| Trees | 3.4 K | 98.79 | | 98.72 | | 87.72 | | 83.42 | |
| Painted metal | 1.3 K | 99.99 | 91.82 | 99.92 | 90.95 | 89.44 | 49.21 | 99.13 | 93.88 |
| Bare soil | 5.1 K | 99.46 | | 98.78 | | 44.99 | | 95.62 | |
| Bitumen | 1.3 K | 99.09 | | 97.66 | | 26.92 | | 87.31 | |
| Bricks | 3.8 K | 97.12 | | 97.12 | | 30.21 | | 98.39 | |
| Shadows | 1 K | 96.62 | | 96.66 | | 93.76 | | 63.02 | |
| Average | - | **92.92** | 91.82 | 92.19 | 90.95 | 57.55 | 49.21 | 88.19 | **93.88** |

### 6.3. Classification Map

Finally, Figure 12 depicts the class map for the Pavia image obtained by the proposed methodology (2D-CNN 1 after post-processing).

Figure 12a describes the true label map and Figure 12b depicts the predicted label map.

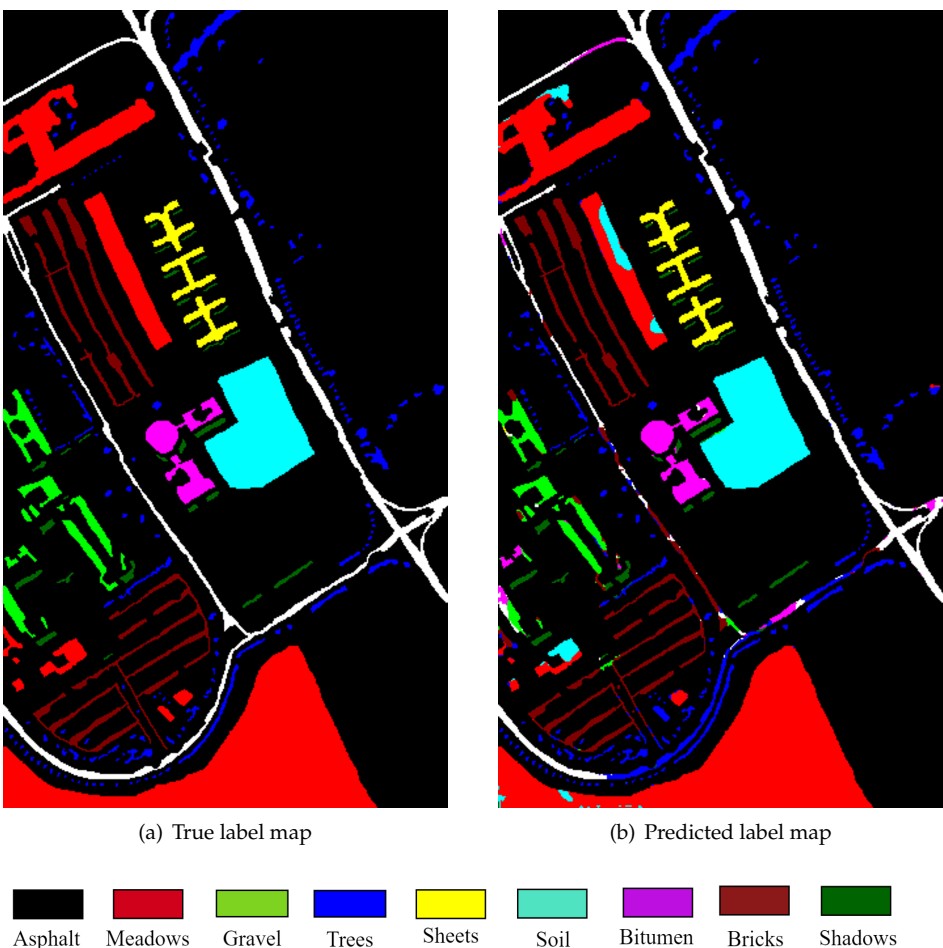

(a) True label map        (b) Predicted label map

Asphalt   Meadows   Gravel   Trees   Sheets   Soil   Bitumen   Bricks   Shadows

**Figure 12.** Classification Pavia image by 2D-CNN 1.

## 7. Conclusions and Future Work

This work proves the suitability of a 2D-CNN-based methodology to deal with satellite images for purposes of crop classification in tropical regions. Specifically, an enhanced 2D-CNN designed into smaller-scale setting, along with a post-processing (able to properly refine the labeling) based on morphological operators results in a recommendable methodology as it reaches competitive results to those reported by recent studies. A remarkable advantage of the proposed methodology was its ability to deal with both imbalanced classes and low-spatial-resolution images.

In addition, it is worth noting that the proposed 2D-CNN architecture may be suitable for other applications related to satellite image analysis, such as the classification of urban materials.

Although the proposed methodology substantially decreases the depth of the 2D-CNN for patch classification purposes and reaches a competitive accuracy, a significant improvement of its overall performance is a matter to be still investigated. Moreover, the exploration of methods such as transfer learning, and generative-adversarial-networks (GANs)-based data augmentation should be an important point in future research to properly address the class imbalance problem.

**Author Contributions:** M.Y.M.-R.: conceptualization, methodology, validation, formal analysis, investigation, resources, writing—original draft preparation, writing—review and editing; L.G.-G.: conceptualization, methodology, investigation, writing—review and editing, visualization, supervision; J.B.G.-M.: conceptualization, methodology, software, validation, investigation, resources, writing—review and editing, visualization, supervision, project administration; J.R.-F.: investigation, resources, writing—review and editing, supervision, project administration, funding acquisition; D.H.P.-O.: methodology, investigation, resources, writing—review and editing, visualization, supervision, project administration, funding acquisition. All authors have read and agreed to the published version of the manuscript'.

**Funding:** This work is supported by the project "Desarrollo de una metodología de visualización interactiva y eficaz de información en Big Data" stated by the Agreement No. 180 1 November 2016 by VIPRI from Universidad de Nariño.

**Institutional Review Board Statement:** Not applicable.

**Informed Consent Statement:** Not applicable.

**Data Availability Statement:** The dataset used in this work can be found publicly available in Internet and is properly cited throughout the manuscript.

**Acknowledgments:** Authors thank to "La fundación Centro de Estudios Interdisciplinarios Básicos y Aplicados (CEIBA)", as well as to SDAS Research Group (www.sdas-group.com).

**Conflicts of Interest:** The authors declare no conflicts of interest.

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
