# Peer review of "Enhanced Convolutional-Neural-Network Architecture for Crop Classification"

_applsci, doi:10.3390/app11094292_

Round 1

Reviewer 1 Report

Automatic crop identification and monitoring is a key aspect in enhancing food production processes, as well as diminishing the related environmental impact. Considering that satellite imagery processing based on deep learning has made significant breakthroughs in different areas, in this work, authors introduce an approach based on multispectral satellite imagery and a 2D-Convolutional-Neural-Network (2D-CNN) architecture to classify 10 different types of crops. The paper is interesting overall, but the following are the comments that must be addressed:

 Comments:

English should be corrected.

  • The authors need to explain what the difference with this study is.

Ji, S., Zhang, C., Xu, A., Shi, Y., & Duan, Y. (2018). 3D convolutional neural networks for crop classification with multi-temporal remote sensing images. Remote Sensing, 10(1), 75.

  • Authors need to elaborate, why the 2D-Convolutional-Neural-Network (2D-CNN) architecture is selected for Crop Classification (Technical reasons must be provided in contrast to state-of-the-art deep learning approaches.
  • Experimental Setup section 6, authors need to present all parameters currently few parameters are missing.
  • Section 1 introduction part is the most important part of the manuscript and authors should need to update the structure of that part that will be good for readers such as AI>>ML>proposed approach with artificial intelligence.

References should be 2018-2021 Web of Science about 50% or more ;30 at least.

Please compare with other methods, justify. Advantages or Disadvantages different methods

Rebetez, J., Satizábal, H. F., Mota, M., Noll, D., Büchi, L., Wendling, M., ... & Burgos, S. (2016, April). Augmenting a convolutional neural network with local histograms-A case study in crop classification from high-resolution UAV imagery. In ESANN. Khan, M.A. and Kim, J., 2020. Toward Developing Efficient Conv-AE-Based Intrusion Detection System Using Heterogeneous Dataset. Electronics, 9(11), p.1771. Mazzia, Vittorio, Aleem Khaliq, and Marcello Chiaberge. "Improvement in land cover and crop classification based on temporal features learning from Sentinel-2 data using recurrent-convolutional neural network (R-CNN)." Applied Sciences 10, no. 1 (2020): 238. Thenmozhi, K., and U. Srinivasulu Reddy. "Crop pest classification based on deep convolutional neural network and transfer learning." Computers and Electronics in Agriculture 164 (2019): 104906.

  • The major contribution of this study looks weak so the authors should need to elaborate more at end of the introduction section.
  • Before Conclusion, please draw a Table and compare with previous researchers, how your approach is better in terms of accuracy. As authors did Table 6. Overall accuracy from Campo Verde database and comparison with [21] and [18]. But at least the previous five papers should be mentioned in that table. So, Table 6 should need to update.
  • Image stacking how authors handle imbalance problem ??? any complexity issue??
  • Conclusion: point out what are you done.

-is there a possibility to use the proposed method for other problems?

- any downside of this study??

Author Response

Thank you very much once more for considering our manuscript ``Enhanced Convolutional-Neural-Network Architecture for Crop Classification'' (referenced as applsci-1163892) by Mónica Moreno, Lorena Guachi, Juan Gómez, Javier Revelo and Diego Peluffo. We have revised the manuscript based on the comments provided by the reviewers. Please find next our replies as well as the corresponding changes (highlighted in blue text) over the manuscript new version. We are grateful for the comments provided and believe that the manuscript has improved greatly based on their feedback. We look forward to hearing whether you find this revision suitable for publication. 

Reviewer 2 Report

Dear Authors,

I found some issues in your work. The most major ones are the following:

1) In line 29 you mention trade-off between computational cost and accuracy as a motivation for this work. However, you never elaborate on how much you gain in computational cost using simpler network. I think you should provide some statistic. 

2) The post-processing does not in my view produce significant improvement. I think you talk too highly of it in the discussion and conclusion parts. It is fine to report small, but not significant improvement.

3) You don't fully utilize the spectral data that you receive from the satellite. 2 dimensional CNN only takes the channels one by one, whereas with 3 dimensional CNN you could analyze the changes each label produce in the spectral dimension. 

Additional notes (typos, oddities, errors and such):

1) Convolutional Neural Network is usually written without the dashes.

2) line 46, bioogical

3) line 79, you have written Langdata baskvist, when you have meant to write Längkvist. You can make ä in latex by \"a

4)Your data is multi-spectral isn't it? Maybe it should be mentioned earlier.

5) line 81, there's a word missing from "as a main conclusion was claimed..." I would add 'it' between 'conclusion' and 'was'

6) Figure 1) In the text you say there are 10 labels, but here I count only 9 labels and two labels that have no representation in data. Clarify.

7) Figure 1) Extra space before period.

8) line 137) 'Firs' -> 'First'

9) Table 1) What is the purpose of the 1x1 convolution in your network?

10) Figure 6) the green part is missing from one of the iterations.

11) line 220) cite Tensorflow. There's an article to be found.

12) line 221) skimage is really called scikit-image. Cite it too

13) lines 228-229) I cant make sense of the sentence here.

14) Eq 4 and 5) Explain also the interpretation of recall and precision

15) Tables 4-6 and 8) Why are some parts highlighted. Explain in the captions.

Best regards,

Author Response

(The authors gave the same response as above.)

Round 2

Reviewer 1 Report

The authors did excellent work to resolve previous queries but this paper still needs to improve.

Authors need to re-write the Abstract in a more meaningful way example (Problem definition=> How existing methods are lacking => proposed solution => Outcome). Also, add accuracy in numbers at end of Abstract

The structure of the paper still looks not good for the readers authors should add these more references in the related work section.

Rebetez, J., Satizábal, H. F., Mota, M., Noll, D., Büchi, L., Wendling, M., ... & Burgos, S. (2016, April). Augmenting a convolutional neural network with local histograms-A case study in crop classification from high-resolution UAV imagery. In ESANN. Khan, M.A. and Kim, J., 2020. Toward Developing Efficient Conv-AE-Based Intrusion Detection System Using Heterogeneous Dataset. Electronics, 9(11), p.1771. Mazzia, Vittorio, Aleem Khaliq, and Marcello Chiaberge. "Improvement in land cover and crop classification based on temporal features learning from Sentinel-2 data using recurrent-convolutional neural network (R-CNN)." Applied Sciences 10, no. 1 (2020): 238. Thenmozhi, K., and U. Srinivasulu Reddy. "Crop pest classification based on deep convolutional neural network and transfer learning." Computers and Electronics in Agriculture 164 (2019): 104906

Conclusion and future work should be revised in a more concert way.

Table 4 and  5. The overall performance should need to update, authors should add training time for each mode.

Figure 2. is the most important for the readers it should be add in the introduction section to authors should add a separate block diagram of the proposed approach in the introduction section.

Image stacking how authors handle imbalance problem? any complexity issue? authors should explain it more technically as given Lin163>> data imbalance affect the overall training of the network and results will be doubtful without data imbalance paresdure>>

Author Response

Thank you very much once more for considering our manuscript ``Enhanced Convolutional-Neural-Network Architecture for Crop Classification'' (referenced as applsci-1163892) by Mónica Moreno, Lorena Guachi, Juan Gómez, Javier Revelo and Diego Peluffo. We have revised the manuscript based on the comments provided by the reviewers. Please find next our replies as well as the corresponding changes (highlighted in a different colors, one per reviewer) over the manuscript's new version. 

Reviewer 2 Report

Dear Authors,

I disagree with your reasoning for not to use 3D-CNN, as it is also useful in cases where no temporal data are considered. It is regularly used in hyperspectral and multispectral studies to analyze the spatial-spectral data, for example in [1,2,3,4,5]. As you do not disclose the amount of Landsat 8 and Sentinel-1 bands you use, I cannot provide my opinion on whether or not it would be useful in your study. Also in the abstract you talk of Modis data, whereas in the text and your source [20] it says Landsat and Sentinel. I wish you would provide a more thorough explination on the used dataset.

Other comments:

1) Choice of RNN as benchmark method seems arbitrary. Do you have any evidence on why it should be useful in this context?

2) in Table 4, should the best f1 result of 2d_cnn-structure 1 be the 92.85 of eukalyptus, or have I misunderstood the table entirely?

3) There are still plenty of typos in the text, especially in the new part. Look it through carefully.

Best regards,

[1] Li, Y.; Zhang, H.; Shen, Q. Spectral–Spatial Classification of Hyperspectral Imagery with 3D Convolutional Neural Network. Remote Sens. 2017, 9, 67. https://doi.org/10.3390/rs9010067 

[2] Z. Ge, G. Cao, X. Li and P. Fu, "Hyperspectral Image Classification Method Based on 2D–3D CNN and Multibranch Feature Fusion," in IEEE Journal of Selected Topics in Applied Earth Observations and Remote Sensing, vol. 13, pp. 5776-5788, 2020, doi: 10.1109/JSTARS.2020.3024841.

[3] Y. Chen et al., "Cloud and Cloud Shadow Detection Based on Multiscale 3D-CNN for High Resolution Multispectral Imagery," in IEEE Access, vol. 8, pp. 16505-16516, 2020, doi: 10.1109/ACCESS.2020.2967590.

[4] Pölönen, Ilkka, et al. "Tree Species Identification Using 3D Spectral Data and 3D Convolutional Neural Network." 2018 9th Workshop on Hyperspectral Image and Signal Processing: Evolution in Remote Sensing (WHISPERS). IEEE, 2018.

[5] Mäyrä, Janne, et al. "Tree species classification from airborne hyperspectral and LiDAR data using 3D convolutional neural networks." Remote Sensing of Environment 256 (2021): 112322.

Author Response

(The authors gave the same response as above.)

Round 3

Reviewer 1 Report

The authors did excellent work and resolve almost previous queries this paper need minor modification as the paper belong to Conv Net so the authors shod add these references that will be good for the readers

Albawi, Saad, Tareq Abed Mohammed, and Saad Al-Zawi. "Understanding of a convolutional neural network." 2017 International Conference on Engineering and Technology (ICET). Ieee, 2017. M.A. and Kim, J., 2020. Toward Developing Efficient Conv-AE-Based Intrusion Detection System Using Heterogeneous Dataset. Electronics, 9(11), p.1771. Xin, R., Zhang, J., & Shao, Y. (2020). Complex network classification with a convolutional neural network. Tsinghua Science and Technology, 25(4), 447-457.

Authors should align all Tables Text with text and all Tables should have the same format, I think Authors should need to check it.

All the equations should gave the same font.

Author Response

Thank you very much once more for considering our manuscript ``Enhanced Convolutional-Neural-Network Architecture for Crop Classification'' (referenced as applsci-1163892) by Mónica Moreno, Lorena Guachi, Juan Gómez, Javier Revelo and Diego Peluffo. We have revised the manuscript based on the comments provided by the reviewers. Please find next our replies as well as the corresponding changes (highlighted in different colors, one per reviewer) over the manuscript's new version. 

Reviewer 2 Report

Dear Authors,

I still spotted some typos here and there. Other than that, I am satisfied with the work. Read the manuscript carefully through once or twice and maybe ask an outsider to proofread it, as one can be blind to their own errors.

Best regards

Author Response

(The authors gave the same response as above.)
